# Potential Activity of Micafungin and Amphotericin B Co-Encapsulated in Nanoemulsion against Systemic *Candida auris* Infection in a Mice Model

**DOI:** 10.3390/jof10040253

**Published:** 2024-03-27

**Authors:** Gabriel Davi Marena, Gabriela Corrêa Carvalho, Alba Ruiz-Gaitán, Giovana Scaramal Onisto, Beatriz Chiari Manzini Bugalho, Letícia Maria Valente Genezini, Maíra Oliveira Dos Santos, Ana Lígia Blanco, Marlus Chorilli, Tais Maria Bauab

**Affiliations:** 1Department of Biological Sciences, School of Pharmaceutical Sciences, São Paulo State University (UNESP), Araraquara 14800-903, Brazil; gabmardavi@gmail.com (G.D.M.); gabrielacarvalho57@yahoo.com (G.C.C.); giovana.scaramal@unesp.br (G.S.O.); beatriz.bugalho@unesp.br (B.C.M.B.); leticia.genezini@unesp.br (L.M.V.G.); maira.oliveira@unesp.br (M.O.D.S.); analigiabl@gmail.com (A.L.B.); 2Department of Drug and Medicines, School of Pharmaceutical Sciences, São Paulo State University (UNESP), Araraquara 14800-903, Brazil; marlus.chorilli@unesp.br; 3Severe Infection Research Group, Health Research Institute La Fe, 46026 Valencia, Spain; 4Department of Medical Microbiology, University and Polytechnic La Fe Hospital, 46026 Valencia, Spain

**Keywords:** *Candida auris*, nanoemulsion, control of infection, nanotechnology strategy, in vivo assay

## Abstract

The *Candida auris* species is a multidrug-resistant yeast capable of causing systemic and lethal infections. Its virulence and increase in outbreaks are a global concern, especially in hospitals where outbreaks are more recurrent. In many cases, monotherapy is not effective, and drug combinations are opted for. However, resistance to antifungals has increased over the years. In view of this, nanoemulsions (NEs) may represent a nanotechnology strategy in the development of new therapeutic alternatives. Therefore, this study developed a co-encapsulated nanoemulsion with amphotericin B (AmB) and micafungin (MICA) (NEMA) for the control of infections caused by *C. auris*. NEs were developed in previous studies. Briefly, the NEs were composed of a mixture of 10% sunflower oil and cholesterol as the oil phase (5:1), 10% Polyoxyethylene (20) cetyl ether (Brij^®^ 58) and soy phosphatidylcholine as surfactant/co-surfactant (2:1), and 80% PBS as the aqueous phase. The in vivo assay used BALB/c mice weighing between 25 and 28 g that were immunosuppressed (CEUA/FCF/CAr n° 29/2021) and infected with *Candida auris* CDC B11903. The in vivo results show the surprising potentiate of the antifungal activity of the co-encapsulated drugs in NE, preventing yeast from causing infection in the lung and thymus. Biochemical assays showed a higher concentration of liver and kidney enzymes under treatment with AmB and MICAmB. In conclusion, this combination of drugs to combat the infection caused by *C. auris* can be considered an efficient therapeutic option, and nanoemulsions contribute to therapeutic potentiate, proving to be a promising new alternative.

## 1. Introduction

Candidemia is known to be the fourth most common bloodstream infection in nosocomial patients. Global agencies are concerned about the high mortality from candidemia, which can be up to 20% in adults. The most common species of candidemia is *Candida albicans*, but new species have emerged as potentially multidrug-resistant pathogens such as *Candida auris* [1].

*C. auris*, a multidrug-resistant yeast, was first reported in the ear canal of a Japanese patient in 2009. The Center for Disease Control and Prevention (CDC) reports that there are five clades of *C. auris* in different regions of the world. Among the justifications for the simultaneous emergence of *C. auris* in different geographic regions are climate changes, easy dissemination, uncontrolled use of antifungals, and prolonged hospitalization times. When isolated and identified, resistance profiles after exposure to the antifungal agent were observed [2,3].

*C. auris* is known for its intrinsic or inducible resistance to the main antifungals available today, such as triazole derivatives and polyenes. Furthermore, several isolates demonstrated in vitro resistance to the two main classes of antifungals, polyenes and echinocandins, representing a clinical and scientific challenge. The resistance of *C. auris* to the main available antifungal agents severely limits treatment options [4].

Unlike other *Candida* species, *C. auris* efficiently colonizes the skin and can contaminate the environment, which may justify the number of outbreaks caused by this pathogen worldwide [5]. Although it is involved in candidemia or invasive infections, the skin is a reservoir for yeast [6]. In the human body, *C. auris* can be found in the nostrils, groin, armpit, and rectum, and can be isolated for 3 months or more after initial detection, even in patients undergoing treatment [7]. Another study points out that the outcome of the infection is serious, with mortality rates close to 70%, mainly due to intrinsic multidrug resistance and the ability to persist on external surfaces and in colonized patients, facilitating human-to-human spreading [8]. Finally, a bibliographical review highlights the ability of *C. auris* to modify the local microbiota with an increase in other potentially pathogenic species under certain circumstances, or even the potential of *C. auris* to move from one region to another and cause a new infectious focus. Therefore, *C. auris* colonization of the skin can be considered a marker of infection, especially in immunocompromised patients sharing the same proximal space [6].

Today, therapeutic options for fungal infections are limited and mainly systemic, contributing to antifungal resistance. Furthermore, contact with antifungals via prophylactic measures and repetitive and long-term treatments for nosocomial are closely correlated with the increase in the occurrence of resistance, making the morbidity and mortality of patients even more serious [9]. In view of this, the need to search for new treatment approaches against these infections is imperative, and it is important for the discovery of new drugs, combinations of antifungals, or tools that allow the therapeutic improvement of the available drugs such as nanoemulsions (NEs). NEs are nanotechnological systems formed by droplets on a nanometric scale. With the aid of surfactants, water and oil form a visibly uniform solution, forming water/oil or oil/water droplets. Therefore, NEs are considered transport tools capable of protecting drugs from degradation, acting in therapeutic potentiation and combating infection more effectively [10,11,12].

The potential of NEs to improve the profile of drugs has been evaluated in several medical applications. The relationship between the surface/volume ratios of NEs contributes to enhancing the absorption and efficacy of incorporated drugs, particularly those drugs that are difficult to dissolve and absorb. The stable interfacial properties of NEs allow drugs to be delivered more selectively and safely to the intended site of action, such as sites of infection. In addition, the NE core serves as an environment to protect the incorporated drugs from environmental changes, including enzymatic degradation, immunological response, and pH changes. These advantages have made NEs a promising alternative in a new therapeutic era, particularly for the development of new therapies against infectious diseases [13].

Combination antifungal therapy can be used to increase the spectrum of activity, increase the potential to kill or inhibit the microorganism, and reduce the risk of developing resistance or toxicity. Therefore, the reason for using combination antifungals is to achieve synergism [14]. This combination is synergistic and can contribute to the enhancement of antifungal activity, overcome drug resistance, and reduce side effects [15]. The combination of two drugs and associates in NE can contribute to infection control since the synergism between two drugs with different mechanisms of action, combined with a biocompatible nanotechnological system, provides greater selectivity to the pathogen and better therapeutic activity [16].

Therefore, considering the serious threat of increasing cases of systemic infection caused by *C. auris* and its high resistance to antifungals treatments (with resistance rates of 93, 35 and 7% to fluconazole, amphotericin B, and echinocandins, respectively), it is imperative that new measures be implemented to control this disease [1,7]. NEs formed by sunflower oil droplets in aqueous solution and stabilized with surfactants can provide a new efficient vehicle for the transport and protection of antifungals and increase therapeutic efficacy against *C. auris*. The NEs developed by sonication showed good stability, uniformity, and desirable encapsulation of AmB and MICA [17,18], which leads to the belief that incorporation of MICA and AmB can improve the antifungal profile in BALB/c mice models with systemic *C. auris* infection. The results obtained in this study can contribute to the hope for a future therapeutic approach based on a nanoemulsion co-encapsulated with drugs already established in the clinical field, and to even avoid possible mechanisms of fungal resistance due to the protective effect that NE can exert. Previous studies have demonstrated the non-toxic profile of this approach in *Galleria mellonella* models, increasing the level of safety for new trials [19]. However, this observation highlights the need for more in-depth studies to be carried out, to increase the reliability of the data.

## 2. Material and Methods

### 2.1. Development and Characterization of NEs

NEs were developed and characterized according to Marena et al. [18,19]. Briefly, the NEs were composed of a mixture of Polyoxyethylene (20) cetyl ether (Brij^®^ 58, Sigma-Aldrich, Steinheim, North Rhine-Westphalia, Germany) and soy phosphatidylcholine (Lipoid, Germany) in a 2:1 ratio (10%), and they were solubilized in 80% Phosphate buffered saline solution (PBS, pH 7.4) followed by the addition of an oily phase composed of sunflower oil (Essential Engineering, São Paulo, Brazil) and cholesterol (Sigma-Aldrich, Germany) in a 5:1 ratio (10%). After the addition of MICA (solubilized in PBS) and AmB (solubilized in sunflower oil), the mixture was sonicated to obtain NEs. MICA (CAS#208538-73-2, CAT#AC-30600) and AmB (CAS# 1397-89-3, CAT# AC-5265) were purchased from APIChem Technology Company (Hangzhou APIChem Technology Co., Ltd., Hangzhou, China).

### 2.2. Strain Fungal

For the execution of in vitro and in vivo assays involving the elucidation of the antifungal potential, the strain of *C. auris* CDC B11903 derived from the Center for Diseases Control (CDC) was obtained commercially from the company Plastlabor, Rio de Janeiro, Brazil. The yeast stock was created from a culture in Sabouraud Dextrose Broth (CSD, BD Difco™, Franklin Lakes, NJ, USA) incubated at 37 °C for 48 h. At the end of the incubation period, aliquots were collected and stored in sterile glycerol (20%) for maintenance at −80 °C. For use, replications were carried out in a culture medium, fresh, and incubated under the previously mentioned conditions.

### 2.3. In Vivo Assay

The in vivo assay was approved by the ethics committee for the use of animals at the Faculty of Pharmaceutical Sciences of Araraquara, Brazil (CEUA/FCF/CAr n° 29/2021). The assay was performed as described by Lepak et al. [20] with modifications. In principle, male mice BALB/c mice (8–10 weeks) with a body weight of 25–28 g were kept under supervision and environmentalization in a reversed cycle (12/12 h) for seven days before the start of the experiment, without water and food restrictions. Three days before infection, the mice received a dose of 200 mg/kg of cyclophosphamide (Sigma-Aldrich, North Rhine-Westphalia, Germany) intravenously and 150 mg/kg after one, three, and five days of infection. This regimen made it possible to render the mice leukopenic (absolute white blood cell count, <100 cells/mm^3^). Mice were infected intraperitoneally with 0.1 mL of sterile saline solution containing 3 × 10^7^ cells/mice. After 4 h of infection, the animals received 0.1 mL of the respective treatments intraperitoneally. The animals were divided into groups of 12 mice to receive the respective treatments and controls: AmB or NEA (0.3 mg/Kg), MICA or NEM (5 mg/Kg), MICAmB or NEMA (0.15 mg/Kg of AmB + 2.5 mg/Kg of MICA), control infection (infection + saline), control without infection (saline only), and control NE (infection + NE). The treatment was carried out every day for 10 days. Between days 1, 4, 7, and 10, three mice/groups were selected for sample collection. The lung, thymus, kidneys, spleen, and liver were collected, washed with saline solution and minced in tubes containing 2 mL of sterile saline until obtaining a homogeneous solution. This solution was cultivated in Sabouraud dextrose agar + chloramphenicol (SDA, BD Difco™, Franklin Lakes, NJ, USA) and incubated at 37 °C for 48 h for quantification of the colony-forming units (CFU/tissue).

### 2.4. Biochemical Analysis

For biochemical tests, blood was collected after decapitation of the mice and transported at a controlled temperature for analysis of glutamic oxaloacetic transaminase (GOT), glutamic pyruvic transaminase (GPT), and creatinine (CRE). The biochemical method was carried out in collaboration with the Laborvet clinical analysis laboratory in Araraquara, São Paulo, Brazil. Quantification was carried out using the Bioclin–Kinetic Methodology using an automated Cobas Mira Plus Roche device (Roche Diagnostic Systems, Indianapolis, IN, USA). All tests were performed in triplicate.

### 2.5. Statistical Analysis

Two-way ANOVA was performed to compare differences between treated and untreated groups, as well as to compare between groups that received treatment; this was followed by a post-Tukey test, with *p* < 0.05. For biochemical tests, one-way ANOVA was performed with *p* < 0.05. For statistical analysis, the GraphPad Prism 8.0 software was used.

## 3. Results

### 3.1. In Vivo Assay

A model of BALB/c mice with systemic infection caused by *C. auris* was created in order to observe the therapeutic efficacy of NEs loaded with MICA and evaluate whether incorporation could interfere, positively or negatively, with the antifungal action of the drugs. Figure 1 shows the results obtained after treatment with AmB and NEA in infected mice; according to the results, it is observed that NEA decreased the fungal burden with greater efficiency when compared to AmB for all organs. The group treated with AmB showed a fungal burden in all organs until the last day of treatment. However, NEA completely fought the infection in all tissues, evidencing the enhancement of the antifungal activity of AmB incorporated in NE. Furthermore, NEA fought the infection in the spleen and thymus after 7 days of treatment (Figure 1B,D). Finally, it is observed that infection in the lung and thymus of mice treated with AmB increased over the days (Figure 1D,E).

On the other hand, Figure 2 show the results obtained during treatment with MICA and NEM in infected mice; according to the results, the group treated with MICA showed better recovery from the infection when compared to the NEM group, since the fungal load in the tissues was higher for mice treated with NEM. However, the fungal burden was significantly higher in the first days of infection, being controlled after 7 days of treatment with NEM. Furthermore, both MICA and NEM prevented the infection from reaching the lung (Figure 2E).

Finally, Figure 3 shows the results obtained during treatment with MICAmB and NEMA; according to the results, the treatment with NEMA was considered the best therapy among the free and encapsulated drugs. NEMA prevented the lung and thymus, organs in the upper region, from being colonized by *C. auris*; this did not occur in groups treated with MICAmB (Figure 3D,E). Three points need to be highlighted: first, NE loaded with the combined drugs increased the antifungal activity; second, the synergism determined by the checkerboard assay was also confirmed in the in vivo assay. The third observation can be evidenced by the better activity of NEMA when compared to NEA (Figure 1) and NEM (Figure 2). In addition to preventing the yeast from infecting the lung and thymus (Figure 3D,E), NEMA controlled the infection after 4, 7, and 7 days of treatment in the liver, spleen, and kidneys, respectively (Figure 3A–C).

### 3.2. Biochemical Analysis

Figure 4 shows the biochemical results of mice infected and treated with free and incorporated drugs; according to the results, treatment in the presence of AmB caused an increase in the concentration of glutamic oxaloacetic transaminase (GOT) and creatinine (CRE). The concentration of GOT in the group treated with AmB was considerably higher when compared to the NEA (*p* = 0.0053) and control (*p* = 0.0155) groups. NEA showed no statistical difference when compared to the control group. MICAmB also showed a considerable increase in GOT concentration compared to NEMA and control (*p* = 0.003 and *p* = 0.002, respectively). Furthermore, the creatinine concentration was significantly higher in the presence of MICAmB compared to the control group (*p* = 0.0035).

## 4. Discussion

Modern medicine, in a way, has brought numerous benefits to human health, such as the treatment of previously incurable infectious diseases and the development of new antimicrobials. However, this medicinal advance has also led to a drastic change in social and political awareness of the dangers that these pathogens can cause, leading to the illusion that everyone is protected from the world of infection. It is evident that these pathogens have adapted to the modern era of medicine, resulting in increased virulence and resistance to treatment. This increase in resistance in recent decades has made some infections intractable. Additionally, lesser-known antimicrobials have emerged at an unprecedented rate. The World Health Organization (WHO) warned of the increase in infections and their rapid global spread in its 2007 report, which was published 17 years ago. Thus, these challenges are a warning to the world of what we can expect in the future, and one of these challenges is the increase of infectious diseases caused by fungi that have long been neglected by public health authorities [21].

The first observation to highlight is the greater presence of fungal loads in the liver, kidneys, and spleen. Abe et al. [22] also found a greater quantity of invasive *C. auris* in gastrointestinal tissue, kidneys, and liver. Torres et al. [23] evaluated the infection profile of *C. auris* strains in neutropenic mice and determined that the organs with the highest fungal load were the kidneys, brain, and heart, with a concentration of 10^5^ cells/g. In the spleen, lung, small intestine, stomach, bladder, and uterus, the fungal load ranged between 10^1^ and 10^3^ CFU/g.

Over the last two decades, there has been a concerning increase in invasive fungal infections, in addition to the increase in fungal infections. What was previously thought to be rare has now become a common reality in hospitals, mainly due to the increase in immunocompromised people. An example is the infection caused by *C. auris*, a previously unknown yeast that is causing great concern worldwide. Clinical isolates show up to 90% resistance to fluconazole, and some clinical isolates are resistant to all classes of currently available antifungals. The mortality rate is impressive and frightening, reaching up to 60%. For this reason, *C. auris* has been classified as an urgent threat to global public health due to its high level of resistance and association with long-term colonization and environmental contamination [24].

To control the new era of resistant microorganisms, it is necessary to develop new therapies that act against these infections. Therefore, nanoemulsions loaded with two important antifungal drugs were developed as a nanotechnological vehicle to reduce systemic *C. auris* infection in immunocompromised mice models.

The NE was previously developed and characterized in our previous studies [17,18,19]. The results showed that the NEs were in a transparent liquid form with 10% oily phase composed of coconut oil, sunflower oil, and cholesterol (5:1); 10% surfactants formed by Brij58 and soya phosphatidylcholine (2:1); and 80% aqueous phase composed of a PBS solution (pH 7.2–7.4). According to physical tests, the NEs had an average size of 40 nm, with a polydispersity index varying between 0.2 and 0.4, indicating a uniform formulation with a slight electronegative charge between −3 mV and −4 mV, according to the zeta potential data. Stability tests showed that the NEs demonstrated stable behaviour for three months at two different storage temperatures: 25 °C and 8 °C. Physicochemical tests showed that the components of NEs, as well as the drugs MICA and AmB, interacted in the formulation, due to the presence of peak shifts in spectrophotometry in the infrared region. Simultaneous thermogravimetry and differential scanning calorimetry (TG-DSC) data also showed interactions between the drug and NE. Finally, morphological tests involving cryogenic transmission electron microscopy (Cryo-TEM) showed that the NEs had a uniform size, exhibiting the spherical structures characteristic of lipid systems such as NEs.

The advantages of NEs in combating infectious diseases have raised hopes of improving the effects of MICA and AmB against *C. auris*, since the evaluation of this combination incorporated into NEs was not found in the literature. Other studies, such as the one carried out by Xavier et al. [25], further increased the possible hypothesis that it could come to fruition. In this study, the authors developed an NE loaded with carvacrol, an antimicrobial monoterpene present in the essential oil extracted from several plants, especially oregano (*Origanum vulgare*). The objective was to increase the effect of the natural product against infection by the *Schistosoma mansoni* parasite in a mouse model. According to the results, NE loaded with carvacrol showed a greater reduction in parasite load (85–90% of parasite inhibition) when compared to free carvacrol (∼30% of parasite inhibition). Finally, the authors concluded that NE can be considered a promising system for controlling schistosomiasis [25].

In view of this, we evaluated the potential of incorporated NEs against systemic *C. auris* infection in a BALB/c mice model; according to the results, NEA and NEMA positively interfered in the antifungal action of the drugs, with a significantly greater reduction in the fungal load in different organs when compared with unincorporated drug therapy. Furthermore, the associated therapy showed a significant reduction in fungal burden, especially NEMA. Treatment with NEMA prevented *C. auris* from causing lung and thymus infection, and the reduction was significantly greater in other organs when compared to the unincorporated association (MICAmB).

The combination of amphotericin B with micafungin in nanoemulsion was not found in the literature, which shows the novelty of this study. However, review research reports the combination of micafungin with amphotericin B and liposomal amphotericin B (LFAB) in patients with pulmonary aspergillosis. According to the review, treatment success was experienced by 26% of patients, who had complete (5%) or partial (20%) responses. The rate of successful patients was 24% for associated therapy and 38% for micafungin monotherapy [14].

Forgács et al. [26] evaluated the antifungal activity of AmB against different clades of *C. auris* in a mouse model; according to the results, AmB increased the survival rate and decreased the burden of yeast in tissues only for groups infected with less virulent strains. Furthermore, the histopathological results confirmed the infiltration of yeasts in the heart, kidneys, and even the central nervous system. According to these results, it is noted that monotherapy was not effective in combating the infection for more virulent strains with less susceptibility to AmB.

In view of this, two drugs co-encapsulated in an NE can help to increase therapeutic action, since two strategies of action can be planned using two drugs with different mechanisms of action; these, combined with the biocompatibility of the system, begin to exert a direct action with low toxic side-effects and a high degree of selectivity. Furthermore, it is important to emphasize that lower doses of the drug can be used. Thus, aiming at a promising alternative (a target for many researchers) to improve the permeability and penetration of drugs is a technological hope for the fight against resistant infections [16,27].

What justifies the better activity of NEMA compared to other treatments is the combination of two drugs with different fungal actions: AmB, added in the oily region of NE, performs action on the membrane via pore formation and cell lysis; MICA, added in the hydrophilic region, acts to inhibit the enzyme 1, 3–β glucan synthase [17,18]. Two drugs with different mechanisms of action present in the same formulation and with the same purpose, reaching the target region efficiently and safely, result in an even more effective action against the infection. Furthermore, one of the components present in the production of the nanoemulsion is cholesterol. Cholesterol has an affinity for the ergosterol present in fungal cells, which possibly favors the selectivity of the nanoemulsion system against yeast [19].

Furthermore, in vivo studies showed that treatment with AmB produces a significant increase in liver enzymes. However, NEs contributed to reducing the concentration of these enzymes, which may suggest a possible positive effect in reducing the toxicity of AmB after incorporation.

The mechanism of action of AmB is to directly bind to ergosterol present in the fungal cell membrane, increase membrane permeability through the formation of aqueous pores, and promote cell lysis and death through the loss of cytoplasmic content. However, administration of AmB is limited due to its infusion-related toxicity profile, an effect postulated to result from the stimulation of pro-inflammatory cytokines. The effect of AmB toxicity is accompanied by nausea, vomiting, hypotension or hypertension, and mainly nephrotoxicity and hepatotoxicity. According to this information, it is believed that the enzyme concentration increased in mice treated with AmB or associated with MICA. For this reason, new alternatives have been sought for the treatment of fungal diseases with the development of lipid structures, such as liposomes, in order to reduce the side effects of AmB. These formulations have been used successfully in therapy, mainly in patients who are intolerant to conventional AmB or who would have difficulty withstanding the action of conventional AmB due to its toxic effects. This reduction in toxicity is related to the better targeting and selectivity of drugs to the target site [18,28,29]. Therefore, the lower enzyme concentration found in the group of mice treated with NEA and NEMA may be related to the benefits provided by lipid systems such as NE.

However, it should be noted that the use of only one strain of *C. auris* is a limiting factor to this study, since today five different clades of *C. auris* have been identified, each with a different resistance profile and infection and virulence mechanisms.

## 5. Conclusions

According to previous studies, NEs present important physical and chemical characteristics, indicating interactions between NEs and drugs. In view of this, analyzing the need to develop new therapies against a multidrug-resistant pathogen (mainly due to the limited therapy available), the therapeutic efficacy of NEs incorporated with amphotericin B and micafungin (as well as in combination) was evaluated. NE increased the antifungal profile of the associated drugs, with a greater decrease in fungal load in different organs. Furthermore, NEMA did not allow the yeast to infect the thymus and lung. Ultimately, NEMA therapy completely fought the infection in shorter intervals. Finally, co-encapsulated NE can contribute to the development of new strategies to control systemic infections caused by *C. auris* and to reduce the emergence of new resistant strains.

## Figures and Tables

**Figure 1 jof-10-00253-f001:**
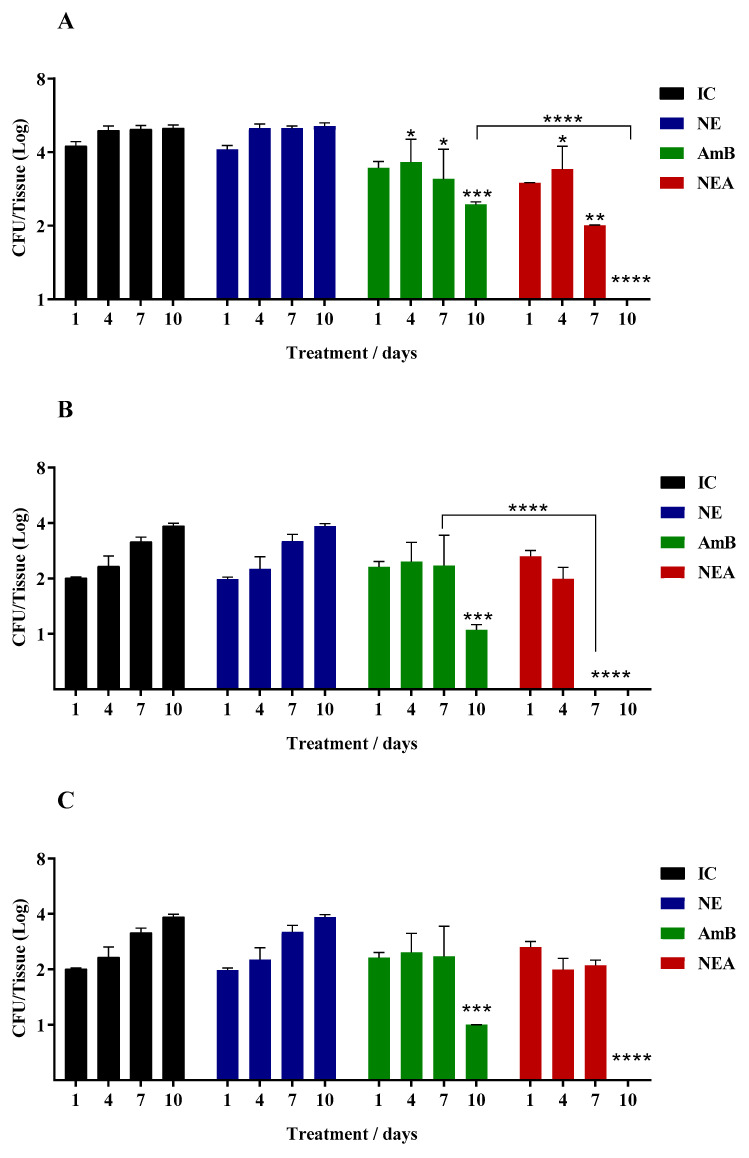
Determination of the antifungal potential of AmB and NEA in mice infected with *C. auris.* Legend: IC: Infection control (infection + saline); NE: control infection with nanoemulsion without the drug (infection + NE); NEA: nanoemulsion + amphotericin B; AmB: amphotericin B. (**A**): liver; (**B**): spleen; (**C**): kidneys; (**D**): thymus; (**E**): lung. *p* < 0.05 (*); *p* < 0.005 (**); *p* < 0.001 (***), *p* < 0.0001 (****).

**Figure 2 jof-10-00253-f002:**
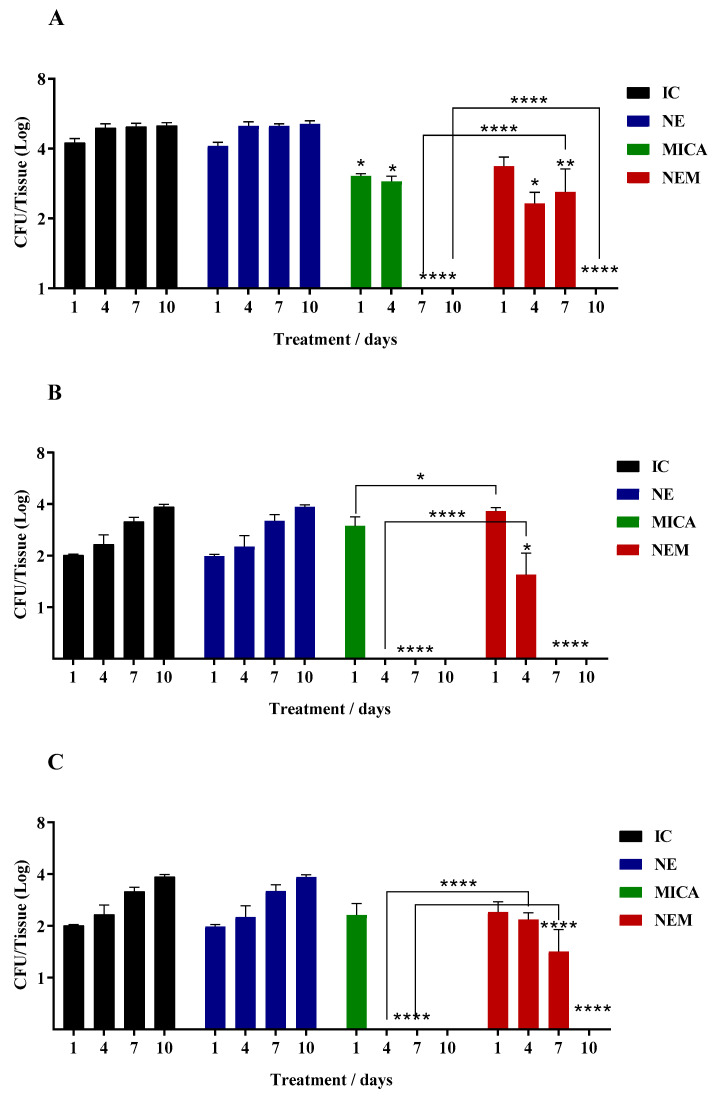
Determination of the antifungal potential of MICA and NEM in mice infected with *C. auris.* Legend: IC: Infection control (infection + saline); NE: control infection with nanoemulsion without the drug (infection + NE); NEM: nanoemulsion + micafungin; MICA: micafungin. (**A**): liver; (**B**): spleen; (**C**): kidneys; (**D**): thymus; (**E**): lung. *p* < 0.05 (*); *p* < 0.005 (**); *p* < 0.0001 (****).

**Figure 3 jof-10-00253-f003:**
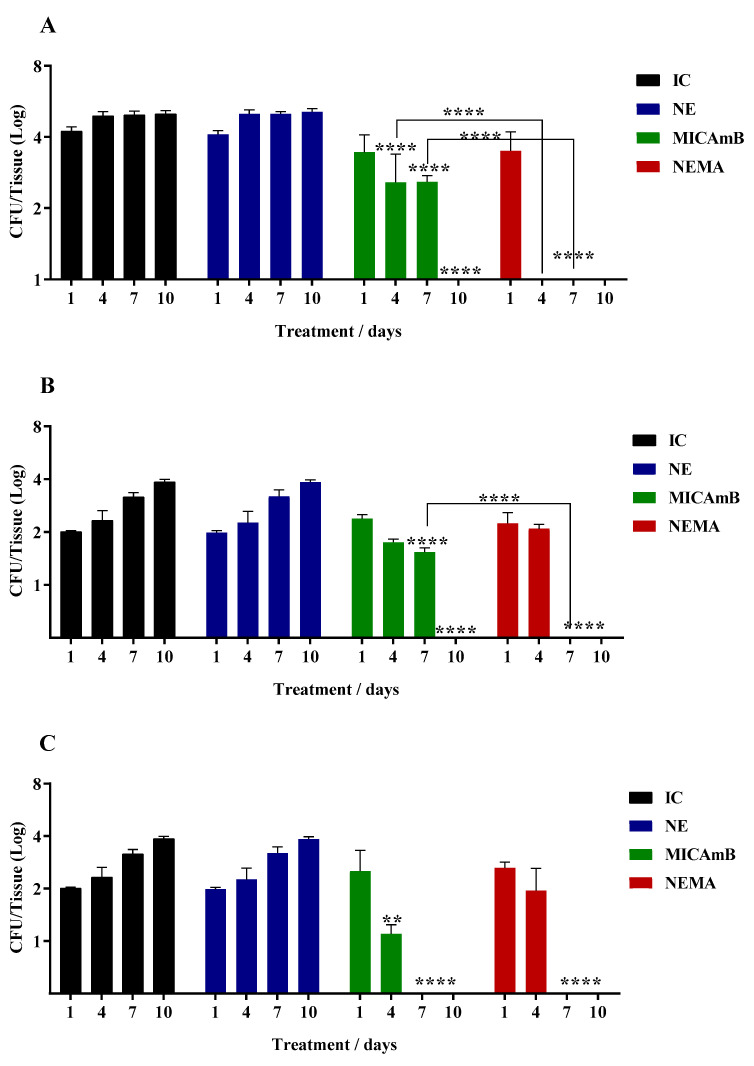
Determination of the antifungal potential of MICAmB and NEMA in mice infected with *C. auris.* Legend: IC: Infection control (infection + saline); NE: control infection with nanoemulsion without the drug (infection + NE); NEMA: nanoemulsion + micafungin + amphotericin B; MICAmB: micafungin + amphotericin B. (**A**): liver; (**B**): spleen; (**C**): kidneys; (**D**): thymus; (**E**): lung. *p* < 0.005 (**); *p* < 0.0001 (****).

**Figure 4 jof-10-00253-f004:**
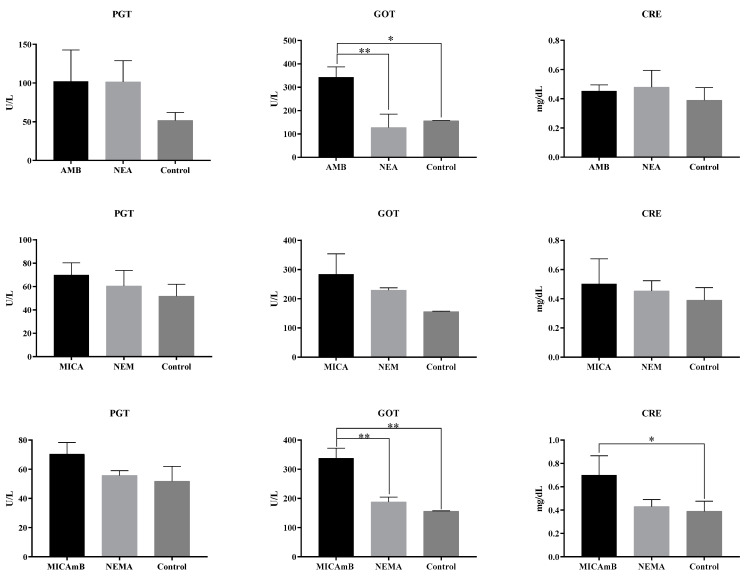
Determination of liver and kidney enzyme concentration in infected and treated mice. Legend: GOT: glutamic oxaloacetic transaminase; PGT: glutamic pyruvic transaminase; CRE: creatinine; NEA: nanoemulsion + amphotericin B; NEM: nanoemulsion + micafungin; NEMA: nanoemulsion + micafungin + amphotericin B; AmB: Amphotericin B; MICA: Micafungin; MICAmB: Micafungin + amphotericin B; control: infected mice without treatment; *p* < 0.05 (*); *p* < 0.005 (**).

## Data Availability

Data are contained within the article.

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
