# Peer review of "Potential Activity of Micafungin and Amphotericin B Co-Encapsulated in Nanoemulsion against Systemic Candida auris Infection in a Mice Model"

_jof, 2024, doi:10.3390/jof10040253_

Round 1

Reviewer 1 Report

Authors submitted a paper entitled “Potential activity of micafungin and amphotericin B co-encapsulated in nanoemulsion against systemic Candida auris infection in a mouse model” for the publication in Journal of Fungi.

The paper has a good scientific soundness and deserves to be published after some revisions.

I suggest adding an abbreviation list, according to the guidelines of this journal.

I have some opinions about the abstract. The abstract lacks information on the specific methodology used in the development and testing of the co-encapsulated nanoemulsion. According to my opinion, it does not mention potential limitations of the study or avenues for future research.

I also have some opinions about the introduction section. The introductory section could benefit from a clear statement of the research objectives or aims of the study. Also, this section lacks a concise overview of the methodology to be employed in the investigation. If authors have little space in the abstract to insight it, they could expand these explanations in this section. Furthermore, it would be helpful to include a brief mention of the significance or potential implications of the study findings in addressing the challenges posed by C. auris infections.

 For specific comments line by line, please follow detail comments section.

Line 178. Please remove double space here.

Line 185. “causing great concern worldwide” please add a reference here.

Line 198. “10%” is this weight percentage on mass basis? Please specifcy.

Regarding the results section:

  1. "indicating good uniformity of the formulation and an electronegative charge" could be revised for clarity, such as: "indicating a uniform formulation with a slight electronegative charge".
  2. "exhibited stable behaviour" can be rephrased as "demonstrated stable behavior".
  3. "carried out by AHGGG" should be clarified with proper citation or explanation. Moreover my advice is to include this inside the abbreviation list.
  4. "free carvacrol (30%)" could benefit from a clearer explanation of the percentage value.
  5. "Therapeutic profile" could be expanded for clarity, for example: "Therapeutic efficacy".
  6. The term "upper organs" could be clarified to avoid confusion. Maybe specify the organs referred to.
  7. Figures mentioned in the text should be labeled consistently and referred to clearly.

Author Response

Reviewer 1

Authors submitted a paper entitled “Potential activity of micafungin and amphotericin B co-encapsulated in nanoemulsion against systemic Candida auris infection in a mouse model” for the publication in Journal of Fungi.

The paper has a good scientific soundness and deserves to be published after some revisions.

- I suggest adding an abbreviation list, according to the guidelines of this journal.

Dear reviewer, thank you for reviewing the manuscript and for the suggestions mentioned.

We added an abbreviation item in the manuscript.

- I have some opinions about the abstract. The abstract lacks information on the specific methodology used in the development and testing of the co-encapsulated nanoemulsion. According to my opinion, it does not mention potential limitations of the study or avenues for future research.

Although we agree with the reviewer, the journal limits the number of words in the abstract. Therefore, we followed your instructions suggested in the following comment.

- I also have some opinions about the introduction section. The introductory section could benefit from a clear statement of the research objectives or aims of the study. Also, this section lacks a concise overview of the methodology to be employed in the investigation. If authors have little space in the abstract to insight it, they could expand these explanations in this section. Furthermore, it would be helpful to include a brief mention of the significance or potential implications of the study findings in addressing the challenges posed by C. auris infections.

We have added this information to the manuscript.

 For specific comments line by line, please follow detail comments section.

Detail comments

- Line 178. Please remove double space here.

OK

- Line 185. “causing great concern worldwide” please add a reference here.

Reference 25 represents the entire paragraph

- Line 198. “10%” is this weight percentage on mass basis? Please specificy.

Yes, 10% of the total weight

Regarding the results section:

  1. "indicating good uniformity of the formulation and an electronegative charge" could be revised for clarity, such as: "indicating a uniform formulation with a slight electronegative charge".

OK

  1. "exhibited stable behaviour" can be rephrased as "demonstrated stable behavior".

OK

  1. "carried out by AHGGG" should be clarified with proper citation or explanation. Moreover my advice is to include this inside the abbreviation list.

Ajusted

  1. "free carvacrol (∼30%)" could benefit from a clearer explanation of the percentage value.

Ajusted

  1. "Therapeutic profile" could be expanded for clarity, for example: "Therapeutic efficacy".

Ok

  1. The term "upper organs" could be clarified to avoid confusion. Maybe specify the organs referred to.

Ok

  1. Figures mentioned in the text should be labeled consistently and referred to clearly.

Ok

Reviewer 2 Report

The manuscript "Potential activity of micafungin and amphotericin B co-encapsulated in nanoemulsion against systemic Candida auris infection in a mouse model" is dealing with co-encapsulated nanoemulsion with amphotericin B and micafungin in the control of infection caused bymulti-drug resistant yeast C. auris. 

The introduction is well written, contains carefully selected references that adequately introduce the topic and issues of the research. The material and work methods are described in detail, which enables the reproducibility of the research. However, the results and discussion are presented a little confusingly. First, the results obtained in this study should be presented, followed by a discussion and comparison with the results of other studies. I suggest that this part be reconstructed in order to see the significance of the obtained results.

The graphics are too small and unreadable.  I suggest that the results be presented in a table with appropriate statistical analysis.

Author Response

Reviewer 2

Major comments

The manuscript "Potential activity of micafungin and amphotericin B co-encapsulated in nanoemulsion against systemic Candida auris infection in a mouse model" is dealing with co-encapsulated nanoemulsion with amphotericin B and micafungin in the control of infection caused by multi-drug resistant yeast C. auris

The introduction is well written, contains carefully selected references that adequately introduce the topic and issues of the research. The material and work methods are described in detail, which enables the reproducibility of the research. However, the results and discussion are presented a little confusingly. First, the results obtained in this study should be presented, followed by a discussion and comparison with the results of other studies. I suggest that this part be reconstructed in order to see the significance of the obtained results.

Dear reviewer, thank you for your attention and suggestions on this manuscript.

We adapted the material as required.

Detail comments

The graphics are too small and unreadable.  I suggest that the results be presented in a table with appropriate statistical analysis.

We believe that the graphs can better represent the data obtained. Therefore, we have improved the quality of the images for better visibility.

Round 2

Reviewer 1 Report

After the first round of revisions, all the major issues have been addressed.

Line 193. "[22][23][24][25][17–19][26]A model of BALB/c mice with" please correct the way these references are written and compact them as [17-19, 11-25] or according to Journal's guidelines.

Author Response

Desculpe, foi um erro de formatação.

Removemos essa informação do manuscrito.

Reviewer 2 Report

The authors have successfully implemented all suggestions. The work has been significantly improved after corrections.

I don't understand the following (numbers in square brackets):

3.1. In vivo assay

[22][23][24][25][17–19][26]A model of BALB/c mice with systemic infection caused by...

[14][27][16,28][20,29][30] 3.2. Biochemical analysis

line 262: [18,31,32]

Please explain or incorporate in the manuscript body

Author Response

Sorry, it was a formatting error.

We removed this information from the manuscript.